

**Projected Changes in Haze Pollution Potential in China**: An
**Ensemble of Regional Climate Model Simulations**
Zhenyu Han[1], Botao Zhou[1,2], Ying Xu[1], Jia Wu[1], Ying Shi[1]
[1] National Climate Center, China Meteorological Administration, Beijing, China
[2] Collaborative Innovation Center on Forecast and Evaluation of Meteorological
Disasters, Nanjing University of Information Science & Technology, Nanjing, China
**Corresponding author**: Botao Zhou
**Corresponding address**: National Climate Center, China Meteorological
Administration, Beijing 100081, China
**E-mail**: zhoubt@cma.gov.cn





**Abstract.** Based on the dynamic downscaling by the regional climate model RegCM4
from three CMIP5 global models under the historical and the RCP4.5 simulations, this
article evaluated the performance of the RegCM4 downscaling simulations on the air
environment carrying capacity (AEC) and weak ventilation days (WVD) in China,
which are applied to measure haze pollution potential. Their changes during the
middle and the end of the 21st century were also projected. The evaluations show that
the RegCM4 downscaling simulations can generally capture the observed features of
the AEC and WVD distributions over the period 1986-2005. The projections indicate
that the annual AEC tends to decrease and the annual WVD tends to increase almost
over the whole country except central China, concurrent with greater change by the
late of the 21st century than by the middle of the 21st century. It suggests that annual
haze pollution potential would be enlarged under the RCP4.5 scenario as compared to
the present. For seasonal change in the four main economic zones of China, it is
projected consistently that there would be a higher probability of haze pollution risk
over the Beijing-Tianjin-Hebei (BTH) region and the Yangtze River Delta (YRD)
region in winter and over the Pearl River Delta (PRD) zone in spring and summer in
the context of the warming scenario. Over Northeast China (NEC), future climate
change might reduce the AEC or increase the WVD throughout the whole year, which
favors the occurrence of haze pollution and thus the haze pollution risk would be
aggravated. Relative contribution of different components related to the AEC change
further indicates that changes of the boundary layer depth and the wind speed play the
leading roles in the AEC change over the BTH and NEC regions. In addition to those



two factors, the precipitation change also exerts dominant impacts on the ACE change
over the YRD and PRD zones.
**Keywords** air environment carrying capacity, ventilation day, haze pollution potential,
regional climate model, evaluation and projection




## 1 Introduction

Haze, as a phenomenon of severe air pollution, exerts remarkably adverse impacts on society and human health, thereby highly concerned by the public and policy makers. Particularly in recent years, heavy haze events hit China frequently (Wang et al., 2014; Zhang et al., 2014) and caused serious damages in many aspects. For instance, they not only increased traffic accidents and delayed traffic (Wu et al., 2005; 2008), but also aggravated ill health problems including respiratory disease, heart disease, cancer and premature death (Wang and Mauzerall, 2006; Xu et al., 2013). Thus, more and more attentions have been paid to the haze pollution in China.

The increasing trend of the haze days in China during recent decades (Ding and Liu, 2014; Song et al., 2014) is documented to be largely attributed to human activities. Due to rapid economic development and urbanization, the pollutants emitted into the atmosphere have been increased, consequently resulting in an intensification of haze pollution in China (Liu and Diamond, 2005; He et al., 2013; Wang et al., 2013b; 2016). Climate change also plays an important role (Jacob and Winner, 2009; Wang et al., 2016). Some studies have indicated that the reduction of surface wind speed, surface relative humidity and precipitation in recent decades (Gao, 2008; Guo et al., 2011; Jiang et al., 2013; Song et al., 2014; Ding and Liu, 2014) provide unfavourable conditions for the sedimentation and diffusion of air pollutants, and thus increase the occurrence of haze pollution in China. Besides, the Arctic sea ice declining under global warming contributes positively to the increase of haze days in eastern China (Wang et al., 2015; Wang and Chen, 2016). Other influential climate



factors for the increase of haze pollution in China, such as the weakening of the East
Asian winter monsoon (Li et al. 2015; Yin et al., 2015) and the northward shifting of
the East Asian jet (Chen and Wang, 2015), are also highlighted. In summary, the
combined effects of increased pollutants and climate change are responsible for the
haze pollution in China.
IPCC AR5 reported that continued emissions of greenhouse gases will cause
further changes in all components of the climate system (IPCC, 2013). From the point
view of the CMIP5 projected change in climate conditions, there are both positive and
negative contributors for the haze pollution in China. For example, the projected
increase in precipitation (Xu and Xu, 2012; Tian et al., 2015; Wu et al., 2015b) is
expected to reduce haze pollution, whereas the decrease of the Arctic sea ice extent
(IPCC, 2013) and the weakening of the East Asian winter monsoon (Wang et al.,
2013a) are inclined to increase haze pollution. So, how the haze pollution in China
will change under the future warming scenario is still an open issue.
Air environment carrying capacity (AEC), which is a combined metric to
measure atmospheric capacity in transporting and diluting pollutants into the
atmosphere, provides a direct way to investigate the change of the haze pollution
potential. When the AEC is low (high), it is unfavourable (favourable) for the
diffusion and cleaning of the pollutants, and thus the haze pollution is (not) prone to
occur. So far, the AEC has been applied in the operation of China Meteorological
Administration (CMA) to forecast haze pollution potential (Kang et al., 2016). On the
other hand, CMIP5 global models show some limitations in simulating regional





climate due to their relatively coarse resolutions (Giorgi et al., 2009). Regional
climate models with higher resolutions are demonstrated to outperform global models
on the regional scale (Lee and Hong 2014; Wu et al. 2015a; Gao et al. 2012, 2016b).
Thus, this study is aimed to project changes of the haze pollution potential in China
from the AEC perspective, based on the downscaling simulations of the regional
climate model RegCM4 under the RCP4.5 scenario.

**2 Model, data and method**
**2.1 Data, regional climate model and simulations**
The regional climate model RegCM4 used in this study is developed by the ICTP
(Giorgi et al., 2012) and applied widely around the world. The model has the
horizontal resolution of 25 km and 18 vertical sigma layers with the top at 50 hPa.
Based on the study of Gao et al. (2016a, b), we selected a suite of physical
parameterization schemes suitable for the simulation of China climate, including the
Emanuel convection scheme (Emanuel, 1991), the radiation package of the CCM3
model for atmospheric radiative transfer (Kiehl et al., 1998), the non-local formulation
of Holtslag (Holtslag et al., 1990) for planetary boundary layer, the SUBEX
parameterization for large-scale precipitation (Pal et al., 2000), and the CLM3.5 for
land surface process (Oleson et al., 2008). The land cover data were updated based on
the vegetation regionalization maps of China (Han et al., 2015).
The domain for the downscaling simulations is the region recommended by
CORDEX-East Asia phase II (Giorgi et al., 2009), covering China continent and





adjacent regions. The RegCM4 simulations, called EC, HAD, and MPI for short, were
driven at 6-hourly intervals by the historical (1979-2005) and RCP4.5 (2006-2099)
simulations from three CMIP5 global models i.e., EC-EARTH, HadGEM2-ES, and
MPI-ESM-MR, respectively. The average of the three simulations with equal weight
is taken as the ensemble mean. The historical simulation denotes the past climate, and
the RCP4.5 represents the medium-low radiative forcing scenario with the radiative
forcing peaking at 4.5 $Wm^{-2}$ by 2100 (Taylor et al., 2012). Readers can visit
http://cmip-pcmdi.llnl.gov/cmip5 for the information about the three CMIP5 models
and the forcing.

To validate the performance of the RegCM4 downscaling simulations, the

ERA-Interim reanalysis dataset (Uppala et al., 2008) with the horizontal resolution of
1.5 °×1.5 °was employed as observations, including 6-hourly boundary layer height,
precipitation, geopotential height and wind speed.
**2.2 Analysis method**

The AEC considers the processes of wet deposition and ventilation and is

expressed in the form:
$$\text{AEC} = C_s \cdot \left( W_r \cdot R \cdot \sqrt{S} + \frac{\sqrt{\pi}}{2} \cdot U_{BL} \cdot H \right) \qquad (1)$$
where $C_s$ is the standard concentration of air pollutant (here, the value is 75 μg m$^{-3}$,
standard concentration for PM$_{2.5}$ in China), $W_r$ is washout constant ($6 \times 10^5$), $R$ is
precipitation, $S$ is unit area and defined as 2500 km$^2$, $U_{BL}$ is mean wind speed
averaged within the boundary layer, $H$ is boundary layer height (Xu and Zhu, 1989).
High (Low) AEC is disadvantageous (advantageous) for the occurrence of haze





pollution, indicating low (high) haze pollution potential. It should be pointed out that
the AEC measures atmospheric carrying capacity in transporting and diluting
pollutants. It does not reflect real emission characteristics. The $C_s$ is the standard
concentration of air pollutant not the real concentration of the pollutant emitted into
the air. For different pollutants, different value can be fixed for $C_s$. Because what we
concerned in this study is the haze pollution potential, its value is set as the standard
concentration for $PM_{2.5}$ in China.

The term $U_{BL} \cdot H$ is named ventilation coefficient (Krishnan and Kunhikrishnan,

2004). Large ventilation coefficient means that a deeper boundary layer can dilute
pollutants and strong winds can remove local pollutants, unfavourable for the haze
occurrence, and vice versa. If each of the 6-hourly ventilation coefficients within one
day is less than 6000 $m^2 s^{-1}$, this day is counted as one weak ventilation day (WVD)
(Leung and Gustafson, 2005). Longer WVD indicates more haze pollution incidents.

According to Eq. (1), the AEC change results from changes in precipitation,

wind speed, and boundary layer depth, which can be simplified as:

$$\Delta AEC = \alpha \cdot \Delta R + \beta \cdot \Delta(U_{BL} \cdot H) \tag{2}$$

where $\alpha = C_s \cdot W_r \cdot R$, $\beta = C_s \cdot \frac{\sqrt{\pi}}{2}$, and $\Delta$ represents the difference between the future and
present-day climate (RCP4.5 minus reference period).

The Eq. (2) could be further decomposed as follows:

$$\Delta AEC = \alpha \cdot \Delta R + \{\beta \cdot \Delta U_{BL} \cdot H_{pd} + \beta \cdot (U_{BL})_{pd} \cdot \Delta H + \beta \cdot \Delta U_{BL} \cdot \Delta H + TR\} \tag{3}$$

The subscript "pd" denotes the present-day climate. The first to third terms in the
right-hand side are associated with changes in precipitation, wind speed within the





boundary layer, and boundary layer depth, respectively. The fourth term is a nonlinear
term including the contribution of changes in both wind speed and boundary layer
depth. Since we use 6-hourly data for the AEC calculation while monthly mean data
for the diagnosis of the change, the last term TR (transient term, deviation from
monthly mean) cannot be ignored, which is obtained as a residual.

The pattern-amplitude projection (PAP) method (Park et al., 2012) is applied to

quantify the relative contributions of individual processes $P_i$ to the AEC change over
certain region.
$$P_i = \frac{\langle \Delta AEC_i \cdot \Delta AEC \rangle}{\langle \Delta AEC \cdot \Delta AEC \rangle} \qquad (4)$$
in which $\langle\ \rangle$ represents area mean, $\Delta AEC_i$ represents components in the
right-hand side of Eq. (3).

**3 Performance of the downscaling simulations**

The performance of the RegCM4 downscaling simulations on the AEC spatial

pattern is firstly evaluated through the comparison with the observation. As shown in
Fig. 1a, the observed AEC is in general large in western China, with the maxima
located over Tibet. Low AEC is found mainly over central and eastern China,
northwestern Xinjiang, and parts of Northeast China. The simulated AEC
distributions from the ensemble (Fig. 1b) and its members (Fig. 1c-e) show general
resemblance to the observation. The spatial correlation coefficients between the
simulation and the observation are all higher than 0.75 (Table 1). On the national
average, the root mean square error (RMES) is small for the ensemble mean and each




member, which varies between 0.47 and 0.54 (Table 1). Nevertheless, there are also
some deficits in the simulations. For example, the AEC is underestimated over the
southern Xinjiang and overestimated over parts of North China.

We further present the observed and simulated distribution of the seasonal AEC

in China during 1986-2005. For the observation, the winter AEC is the lowest among
the four seasons in a broad region of China (Fig. 2a). In spring, the AEC increases
significantly and the regions with high AEC expand obviously. The central eastern
China is dominated by the low capacity (Fig. 2c). Compared with the case in spring,
the summer AEC increases over central China while decreases slightly over Tibet and
Northeast China (Fig. 2e). The AEC distribution in autumn is similar to that in winter
but with larger capacity over the regions except Tibet (Fig. 2g). The seasonal
variation of the AEC in the ensemble simulation agrees with that in the observation
although there are some discrepancies (Fig. 2b, 2d, 2f and 2h). The spatial correlation
coefficient between the simulation and the observation ranges from 0.61 to 0.79 and
the RMES is in the range of 0.47 to 0.76 for the national average in four seasons
(Table 2).

The WVD distribution during 1986-2005 in the observation and the ensemble

simulation is displayed in Fig. 3a and Fig.3b, respectively. It is noticed that the
simulated pattern and the observed pattern are approximate to each other. Namely, the
number of weak ventilation days per year is relatively small over Tibet while
relatively large over central and eastern China, Northeast China, southern North



China and Xinjiang. The spatial correlation between them is 0.74. However, we also
note that the WVD is overestimated by the ensemble simulation.

The wet deposition is observed to be large over southern China and the south

edge of the Tibetan Plateau while small over northwestern China (Fig. 3c). According
to Eq. (1), the wet deposition pattern exactly corresponds to the distribution of
precipitation. The observed features can also be captured by the ensemble simulation
(Fig. 3d). The spatial correlation coefficient between the simulation and the
observation is up to 0.84.

In brief, the downscaling simulations of the RegCM4 can reasonably reproduce

the observed characteristics of the distribution of the AEC, WVC and wet deposition
in China. It provides justification to use them for the future projection.

**4 Projected changes**

Fig. 4 exhibits the ensemble projected changes in AEC, WVC and wet deposition

during the middle of the 21st century (2046-2065) and the end of the 21st century
(2080-2099) relative to the reference period 1986-2005. A general decrease in AEC
and an overall increase in WVC are projected over almost the whole country except
central China in the context of the RCP4.5 scenario. The change in magnitude is
larger by the end of the 21st century than by the middle of the 21st century. The
maximum decrease in AEC appears at the edge of the Qinghai-Tibet Plateau and the
Loess Plateau, with the percentage change being 4% for the middle of the 21st
century and 5% for the end of the 21st century. The relatively large decreases are





located in Southwest China, northern North China, Northeast China and Inner
Mongolia (Fig. 4a and Fig.4b). The increase in WVD is projected to be particularly
pronounced in western and northern China (Fig. 4c and Fig. 4d). The three ensemble
samples agree well on the sign of the changes, indicative of a good consistency in the
projection. In contrast, there would be an increasing tendency for the AEC and a
decreasing tendency for the WVD over central China where the climatological
capacity is low in the reference period 1986-2005. However, the sign of the projected
change is inconsistent among the three ensemble samples. Compared with the
ensemble projection, the EC and HAD show relatively large discrepancy for the sign
of the projected change in AEC and WVD, respectively (Figures not shown).
For the change in wet deposition, a general increase is projected across China,
also with greater change in 2080-2099 than in 2046-2065 (Fig. 4e and Fig. 4f). In
addition, we can find inconsistent signs of the projected change over southern China
during 2046-2065 (Fig. 4e) and over some parts of Northeast China during 2080-2099
(Fig. 4f). The inconsistent during 2046-2065 (2080-2099) is mainly due to the
difference of the HAD (MPI) projection from the other two ensemble members
(Figures not shown).
Following, we turn to examine the seasonal and annual changes of the AEC and
WVD over the four main economic zones of China which suffer severely from the
haze pollution at present, i.e., Beijing-Tianjin-Hebei region (BTH), Northeast China
(NEC), Yangtze River Delta economic zone (YRD), and Pearl River Delta economic
zone (PRD) in more detail.

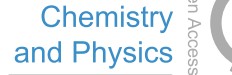

1) Beijing-Tianjin-Hebei region

As shown in Fig. 5a, the ensemble projection indicates a decrease of the AEC in

all four seasons during the middle of the 21st century. The percentage change relative
to 1986-2005 is the lowest in spring and the largest in winter. The changes in summer
and autumn are between -2% and -3%. The three ensemble members agree on the sign
of the changes in all seasons except spring but with different spread. For the summer
season, the spread is the smallest. While in other seasons, it is close to or larger than
the ensemble projected change. During the end of the 21st century, the decrease of the
AEC is further enhanced, with the largest enhancement occurring in winter. Moreover,
the spread in general becomes much larger. For annual change, both the ensemble and
its members project that the AEC would reduce during the middle and the end of the
21st century with the larger amplitude in the latter period.

As for the WVD (Fig. 5b), an increasing tendency is projected by the ensemble

for annual and seasonal mean during the middle of the 21st century. The change is the
smallest in summer and the largest in winter. The ensemble members show good
agreement on the positive change in winter, autumn, and annual mean. During the late
of the 21st century, the increase in WVD is further enlarged in winter and autumn
while it is reduced in spring and summer. There is no appreciable change for annual
mean as compared to that in the middle of the 21st century. Only for the winter season
and annual mean, all the individual simulations consistently show the same projection
as the ensemble.



2) Northeast China

The annual and seasonal AEC is projected by the ensemble to decrease during

the middle of the 21st century, and the percentage changes are comparable among
four seasons and annual mean (Fig. 6a). The ensemble members also project negative
tendency consistently except in spring. Compared with the middle of the 21st century,
the case for the end of the 21st century is similar but with larger decrease. Besides, all
the three ensemble members show good consistence for the projection.

The WVD is projected by the ensemble and its members to increase during the

middle and the end of the 21st century for annual mean and all four seasons (Fig. 6b).
Similarly, the projected change is larger during the end of the 21st century than during
the middle of the 21st century, with the largest increase appearing in spring.
3) Yangtze River Delta economic zone

The ensemble projection indicates that the AEC would decrease for annual mean

and all the seasons except autumn (Fig. 7a). The percentage change is the smallest in
spring (with the decrease of less than 1%) and the greatest in winter (with the
decrease of more than 3%). The counterparts for summer and autumn are about -2%
and 1%, respectively. However, large spread exists among the projections of the three
ensemble members. Only for winter and annual mean, they project the same sign of
the change. At the end of the 21st century in the ensemble projection, the decrease in
AEC is enhanced to 6% in winter. Consistent change is projected by the ensemble
members. In contrast, the decrease in summer and the increase in autumn are
weakened as compared to the middle of the 21st century. A slight increase of the AEC



is found in spring. For annual mean AEC, the decrease is somewhat larger by the end
of the 21st century than by the middle of the 21st century.
The WVD for annual mean, winter and spring is projected by the ensemble to
increase, with larger change during the end of the 21st century than during the middle
of the 21st century (Fig. 7b). The greatest change occurs in winter. For summer, the
ensemble projects that the WVD almost remains unchanged during the middle of the
21st century while increases at the end of the 21st century. For autumn, the ensemble
projects that the WVD decreases slightly during the middle of the 21st century while
increases slightly by the end of the 21st century. The ensemble members show good
consistency of the projections for winter and annual mean during both periods.
4) Pearl River Delta economic zone
As projected by the ensemble (Fig. 8a), the annual, spring and summer AEC
would decrease. Such a decrease is relatively larger during the middle of the 21st
century than during the end of the 21st century and the greatest decrease occurs in
spring. For winter, the AEC is projected to increase and be comparable during the
middle and the end of the 21st century. For autumn, the projected AEC decreases by
about 1% over the period 2046-2065 and increase by about 0.5% over the period
2080-2099. However, the projections from the three members are not consistent for
all four seasons.
The ensemble projects an increase in WVD for annual mean and four seasons,
with the greatest increase in summer during the middle of the 21st century (Fig. 8b).
The individual members consistently show the positive change for spring, summer,





and annual mean. Compared with the middle of the 21st century, the increase of the
WVD is reduced in summer while enhanced for annual mean and the remaining
seasons during the late of the 21st century. The autumn is the season with the
maximum change. The individual members show the same projections as the
ensemble on the sign of change still for spring, summer, and annual mean.

The consistence of the three ensemble members on the direction of the projected

change which can be used to visualize the uncertainty in the projection is further
summarized in Table 3. In general, although there are some uncertainties on the
regional changes, the three members consistently project a decrease of the AEC or an
increase of WVD for annual mean over the four economic zones, especially over the
Beijing-Tianjin-Hebei region and Northeast China. It signifies that future climate
change will contribute positively to the haze pollution in these regions. For seasonal
change, decrease in AEC or increase in WVD, is projected consistently to appear in
all four seasons over Northeast China. It suggests that there would be an increase of
haze pollution potential throughout the whole year. Besides, the consistent projections
indicate a higher potential risk of haze pollution over the Beijing-Tianjin-Hebei
region and the Yangtze River Delta region in winter and over the Pearl River Delta
zone in spring and summer.

The temporal evolution of the annual and seasonal AEC and WVD over the four

main economic zones are also plotted (Figs. 5-8 c-g), and the corresponding trend
values projected by the ensemble for the period of 2016-2099 are summarized in
Table 4. Theil-Sen trend analysis (Theil, 1950; Sen, 1968) was used to estimate the





trends and the non-parametric Mann-Kendall test (Mann, 1945; Kendall, 1975) was
used for significant test. Generally, the secular variations of the AEC and the WVD
show some diversity across different seasons over the regions except NEC where a
decrease in AEC and an increase in VWD is projected uniformly. Nevertheless, for
the trends significant above the 95% level, it is interesting to notice that the decrease
in AEC is mostly accompanied with the increase in WVD, for instance for winter over
TBH, for annual mean and all the seasons over NEC, for annual mean, winter and
summer over YRD, and for annual mean and autumn over PRD.

**5 Contributions of different factors to the change of AEC**
Based on Eqs. (2) and (3), we further investigate the contribution of different
factors to the projected change in AEC. For brevity, we only show the results for the
period 2046-2065 in the following, because the case for the period 2080-2099 is
similar.
Figs. 9a and 9b exhibit relative contributions to the annual AEC change over the
course of 2046-2065 from changes in precipitation and ventilation, respectively.
Overall, the ventilation change plays a dominant role in and contributes positively to
the change of the AEC over most parts of China, particularly in western and northern
China (Fig. 9b). In contrast, the relative contribution of the precipitation change is in
general negative over western and northern China while positive over southern China
(Fig. 9a).

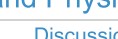

According to Eq. (3), the effect of ventilation change can be decomposed into
four terms, i.e., wind speed change, boundary layer depth change, nonlinear term, and
transient term. Among these contributors for annual ventilation change, the effects of
boundary layer depth (Fig.9d) and wind speed (Fig.9c) are relatively large and the
former is greater than the latter over most parts of eastern China. The transient term
also exert effects for instance over some parts of western and southern China (Fig.9f),
while the effects of the nonlinear term are tiny across China (Fig. 9e).
Fig. 10 further presents relative contributions of aforementioned factors to
annual and seasonal AEC change over the four economic zones as projected by the
ensemble and its members. As shown in Figs. 10a and 10b, changes in wind speed
and boundary layer depth have the greatest contributions to the AEC change over the
THB and NEC regions for annual mean and all the seasons except summer. The
contribution from the precipitation is in general relatively small. Besides, the effects
of the transient term are larger than that of the precipitation, and the effects of the
nonlinear term can be negligible. These results indicate that changes in wind speed
and boundary layer depth are the leading contributors responsible for the AEC change
over the two regions. In contrast, over the YRD (Fig.10c) and PRD (Fig.10d) zones,
change in precipitation also plays a dominant role. The contribution from the
precipitation change is comparable to and even larger than that from changes in wind
speed and boundary layer depth for all the seasons except winter.



**6 Conclusion**

In this study, we conducted downscaling simulations by use of the RegCM4 driven by three CMIP5 models' results under the historical simulation and the RCP4.5 scenario. On this basis, we evaluated the fidelity of the RegCM4 simulations on the AEC and WVD which are indictors for haze pollution potential, and then projected their change during the middle and the end of this century for China and four main economic zones. The major findings are summarized below:

1) The evaluation indicates that the RegCM4 downscaling simulations in general show good performance in modeling the climatological distribution of the annual and seasonal AEC, despite some discrepancies in certain regions. The spatial correlations between the simulation and the observation for annual mean and four seasons are higher above 0.6. The simulations also well capture the observed WVD pattern with relatively small WVD over Tibet and relatively large WVD over central and eastern China, Northeast China, southern North China and Xinjiang, although the WVD is overestimated systematically.

2) The annual AEC and WVD are respectively projected by the ensemble to decrease and increase almost in the entire region except central China, accompanied with larger amplitude by the end of the 21st century than by the middle of the 21st century. The decreases in AEC are relatively large over Tibet, Southwest China, northern North China, Northeast China and Inner Mongolia. The increase in WVD is particularly pronounced in northern China. The individual members present consistent projections of changes as the ensemble. In contrast, the ensemble projects an increase



in AEC and a decrease in WVD over central China. However, the sign of the
projected change is inconsistent among the ensemble samples.

3) The consistency analysis suggests that there would be a high probability of the

increase in air pollution risk over the BTH and YRD regions in winter and over the
PRD zone in spring and summer in a warmer world. Over NEC, climate change might
reduce the AEC or increase the WVD throughout the whole year, favorable for the
occurrence of haze pollution and also indicative of an aggravation of haze pollution
risk. Furthermore, the contribution analysis indicates that changes in boundary layer
depth and wind speed play the leading roles in the AEC change over the BTH and
NEC regions. In addition to the aforementioned two factors, the precipitation change
is also a dominant factor influencing the ACE change over the YRD and PRD zones.

In this study, we mainly showed the downscaled results driven by three global

models. Note that the planetary boundary layer depth is not a standard CMIP5 output
variable, and the coarse vertical resolution of the global models prevents us from
estimating the planetary boundary layer depth. Moreover, the CMIP5 experiments did
not supply high-frequency (six-hourly) outputs for calculating AEC and WVD. These
make it hard to estimate whether the consistencies and inconsistencies of the
projection is caused by the global models or to some extent affected by the dynamical
downscaling of the regional model.





**Acknowledgments.** This research was jointly supported by the National Key
Research and Development Program of China (2016YFA0600701), the National
Natural Science Foundation of China (41675069, 41405101), and the Climate Change
Specific Fund of China (CCSF201626).



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

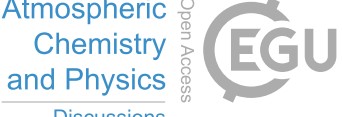

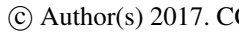


**Captions:**
**Table 1.** Statistic results for the simulation skills in annual mean AEC for the period
of 1986-2005.
**Table 2.** Statistic results for the ensemble simulation skills in seasonal AEC for the
period of 1986-2005.
**Table 3.** The consistence of the three ensemble members on the direction of the
projected change over the four economic zones of China. Consistent projection
on the decrease in AEC is markerd by √ and that on the increase in WVD is
marked by ☆.
**Table 4.** Trends of AEC and WVD (%/10a) over the four economic zones of China,
based on 9-year running mean time series of the percentage change during
2016-2099. Asterisks indicate the trends are statistically significant above the 95%
confidence level.
**Figure 1.** Spatial distribution of annual AEC (unit:$10^4$t/a/km) during 1986-2005: (a)
observation, (b) ensemble, (c) EC, (d) HAD, (e) MPI.
**Figure 2**. Spatial distribution of seasonal AEC (unit: $10^4$t/a/km) during 1986-2005:
(a-b) winter, (c-d) spring, (e-f) summer, (g-h) autumn. Left panel is for the
observation and the right panel is for the ensemble simulation.
**Figure 3.** Spatial distribution of (a-b) the number of weak ventilation days per year
and (c-d) wet deposition (unit: $10^4$t/a/km) during 1986-2005: (a, c) observation,
(b, d) ensemble simulation.





**Figure 4.** Ensemble projected percentage changes (relative to 1986-2005) in (a-b)
AEC and (c-d) WVD during (a, c) 2046-2065 and (b, d) 2080-2099. Hatched
regions indicate all ensemble members agree on the sign of change.
**Figure 5.** Range of projected percentage changes (relative to 1986-2005) in (a) AEC
and (b) WVD during 2046-2065 and 2080-2099, and 9a running mean time series
of percentage changes in (c) annual, (d) winter (DJF), (e) spring (MAM), (f)
summer (JJA), (g) autumn (SON) for the Beijing-Tianjin-Hebei region. In Figure
(a-b), the bars represent the ensemble projection and the marks represent the
individual projection of the three members; the left (right) bar in each group is for
2046-2065 (2080-2099). In Figure (c-g), the solid (dashed) lines represent
changes in AEC (WVD).
**Figure 6.** Same as Figure 5, but for Northeast China.
**Figure 7.** Same as Figure 5, but for Yangtze River Delta economic zone.
**Figure 8.** Same as Figure 5, but for Pearl River Delta economic zone.
**Figure 9.** Relative contributions (unit: %) of individual components to annual AEC
change in the middle of the 21st century based on the ensemble results. (a)
precipitation, (b) ventilation, (c) wind speed averaged with the boundary layer, (d)
boundary layer depth, (e) nolinear term, (f) transient term.
**Figure 10.** Relative contributions (unit: %) of individual components to annual AEC
change in the middle of the 21st century averaged over four main economic
zones of China: (a) BTH, (b) NEC, (c) YRD, (d) PRD. The bars represent the
ensemble projection and the marks represent the individual projection of the three





members. Bars from left to right in each group are in turn for annual, DJF, MAM,
JJA, and SON.





**Table 1.**  Statistic results for the simulation skills in annual mean AEC for the period

of 1986-2005.

| Simulations | Pattern correlation coefficient (CC) | Root mean square error (RMES) |
|---|---|---|
| EC | 0.76 | 0.47 |
| HAD | 0.79 | 0.54 |
| MPI | 0.75 | 0.48 |
| Ensemble | 0.77 | 0.49 |






**Table 2.**   Statistic results for the ensemble simulation skills in seasonal AEC for the

period of 1986-2005.

| Season | Pattern correlation coefficient (CC) | Root mean square error (RMES) |
|--------|--------------------------------------|-------------------------------|
| Winter | 0.79 | 0.76 |
| Spring | 0.75 | 0.68 |
| Summer | 0.61 | 0.56 |
| Autumn | 0.78 | 0.47 |






**Table 3.** The consistence of the three ensemble members on the direction of the
projected change over the four economic zones of China. Consistent projection on the
decrease in AEC is markerd by √ and that on the increase in WVD is marked by
☆.

| Economic zone | Period | ANN | DJF | MAM | JJA | SON |
|---|---|---|---|---|---|---|
| BTH | 2046-2065 | √☆ | √☆ | | √ | √☆ |
| | 2080-2099 | √☆ | ☆ | | | |
| NEC | 2046-2065 | √☆ | √☆ | ☆ | √☆ | √☆ |
| | 2080-2099 | √☆ | √☆ | √☆ | √☆ | √☆ |
| YRD | 2046-2065 | √☆ | √☆ | | | |
| | 2080-2099 | ☆ | √☆ | | | |
| PRD | 2046-2065 | ☆ | | ☆ | ☆ | |
| | 2080-2099 | ☆ | | ☆ | ☆ | |






**Table 4.** Trends of AEC and WVD (%/10a) over the four economic zones of China,
based on 9-year running mean time series of the percentage change during 2016-2099.
Asterisks indicate the trends are statistically significant above the 95% confidence
level.

| Economic zone | Variable | ANN | DJF | MAM | JJA | SON |
|---|---|---|---|---|---|---|
| BTH | AEC | -0.41* | -0.96* | 0.02 | -0.19* | -0.80* |
| | WVD | 0.33 | 2.30* | -1.53* | -0.51* | 0.55 |
| NEC | AEC | -0.46* | -0.76* | -0.26* | -0.41* | -0.61* |
| | WVD | 1.49* | 2.60* | 1.30* | 0.73* | 0.97* |
| YRD | AEC | -0.27* | -1.17* | 0.32* | -0.45* | -0.02 |
| | WVD | 0.51* | 0.88* | -0.26 | 0.71* | -0.15 |
| PRD | AEC | -0.14* | -0.03 | -0.22* | -0.12 | -0.29* |
| | WVD | 1.17* | -0.01 | -0.30 | 2.17* | 1.50* |




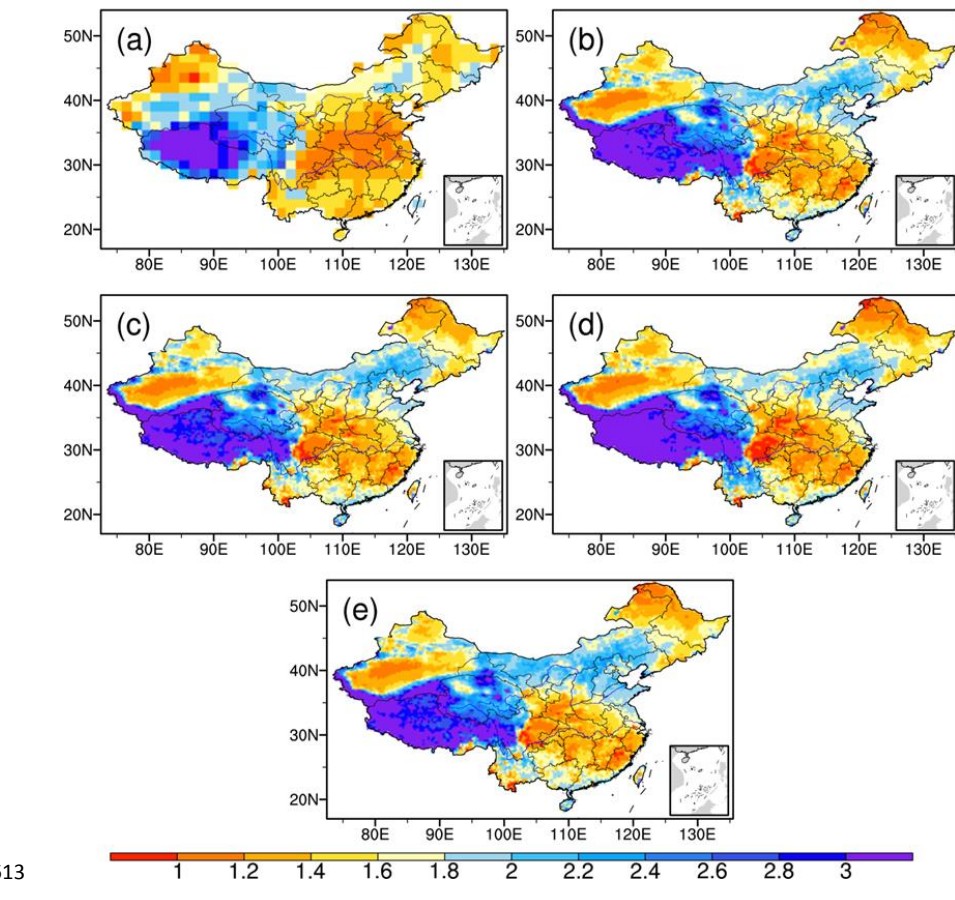


**Figure 1.** Spatial distribution of annual AEC (unit:$10^4$t/a/km) during 1986-2005: (a)

observation, (b) ensemble, (c) EC, (d) HAD, (e) MPI.





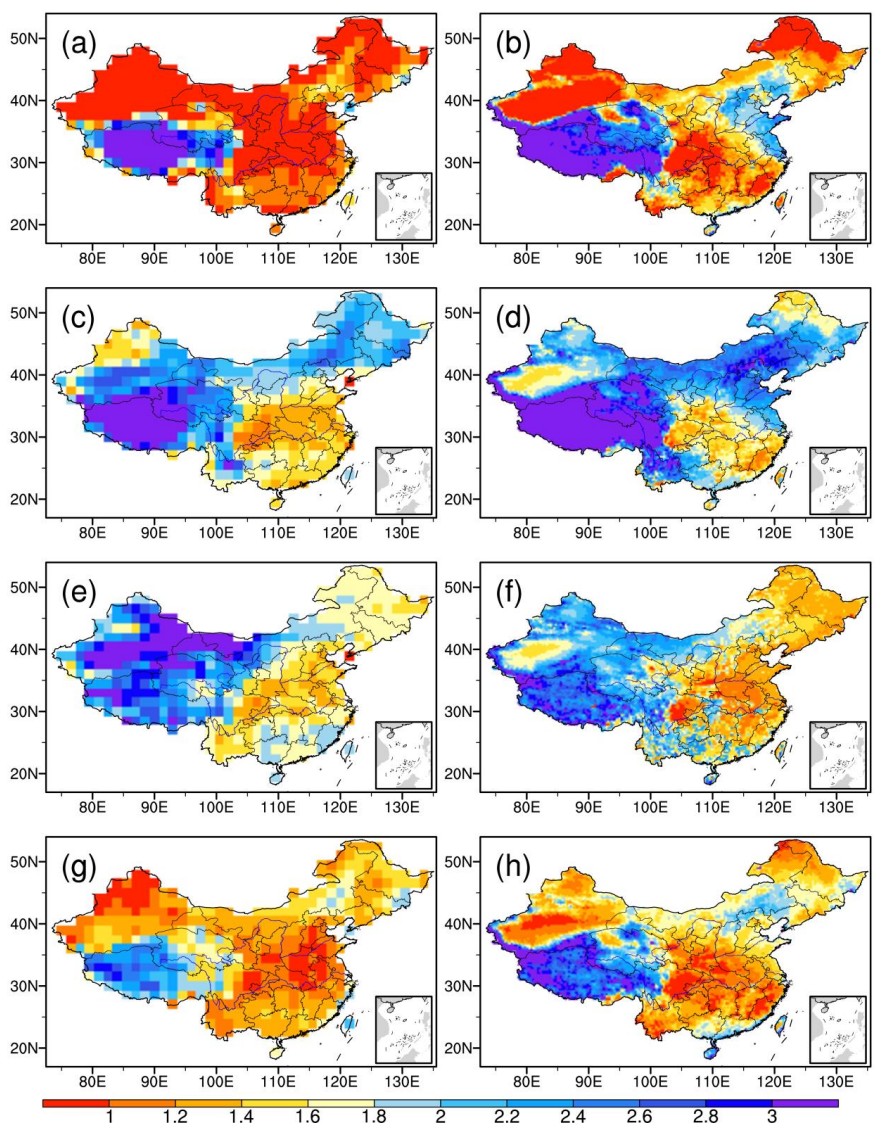

**Figure 2**. Spatial distribution of seasonal AEC (unit: $10^4$t/a/km) during 1986-2005:

(a-b) winter, (c-d) spring, (e-f) summer, (g-h) autumn. Left panel is for the

observation and the right panel is for the ensemble simulation.





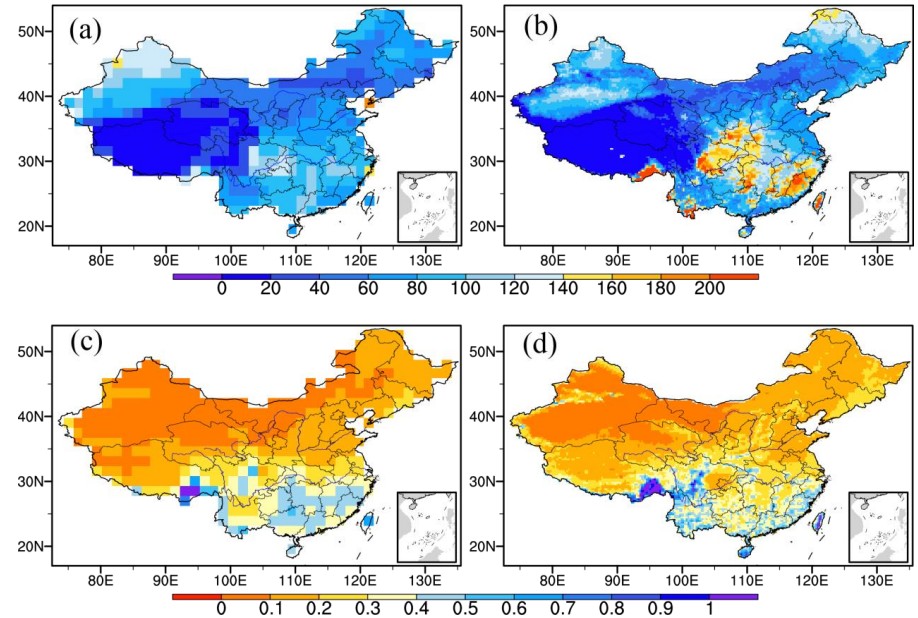

**Figure 3.** Spatial distribution of (a-b) the number of weak ventilation days per year
and (c-d) wet deposition (unit: $10^4$t/a/km) during 1986-2005: (a, c) observation, (b, d)
ensemble simulation.





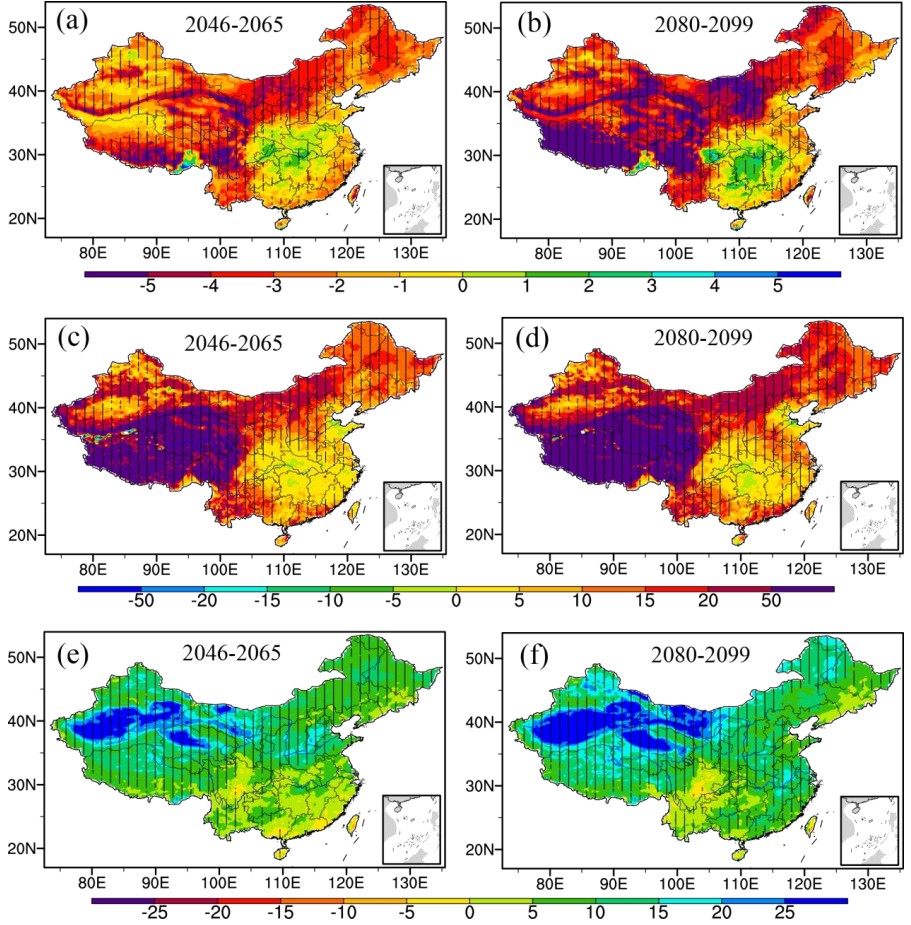

**Figure 4.** Ensemble projected percentage changes (relative to 1986-2005) in (a-b)

AEC, (c-d) WVD, and (e-f) wet deposition during 2046-2065 (left panel) and

2080-2099 (right panel). Hatched regions indicate all ensemble members agree on the

sign of change.





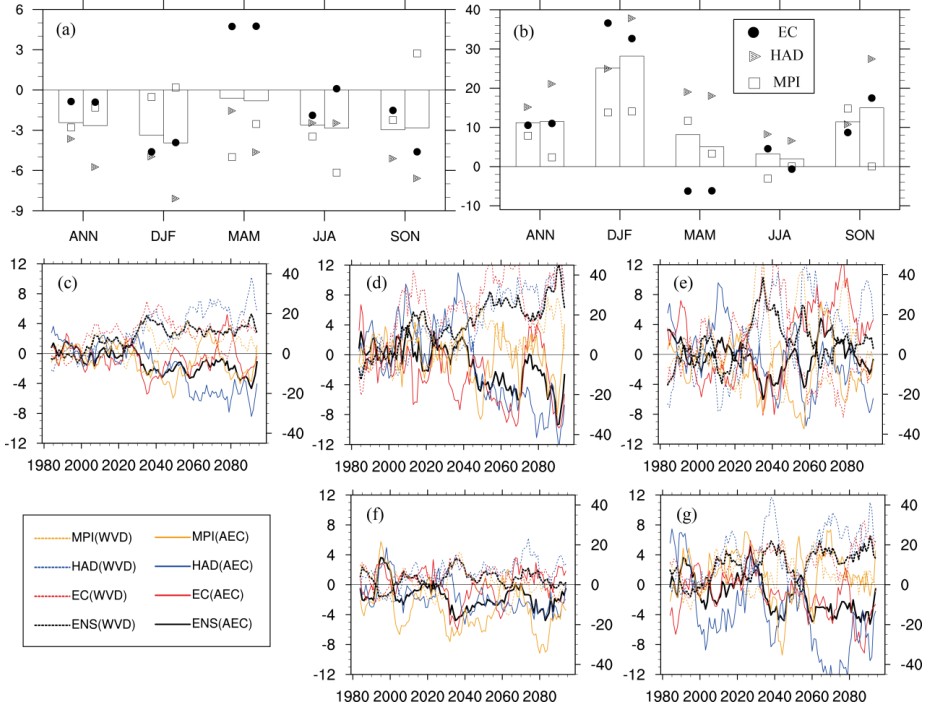

**Figure 5.** Range of projected percentage changes (relative to 1986-2005) in (a) AEC

and (b) WVD during 2046-2065 and 2080-2099, and 9a running mean time series of

percentage changes in (c) annual, (d) winter (DJF), (e) spring (MAM), (f) summer

(JJA), (g) autumn (SON) for the Beijing-Tianjin-Hebei region. In Figure (a-b), the

bars represent the ensemble projection and the marks represent the individual

projection of the three members; the left (right) bar in each group is for 2046-2065

(2080-2099). In Figure (c-g), the solid (dashed) lines represent changes in AEC

(WVD).





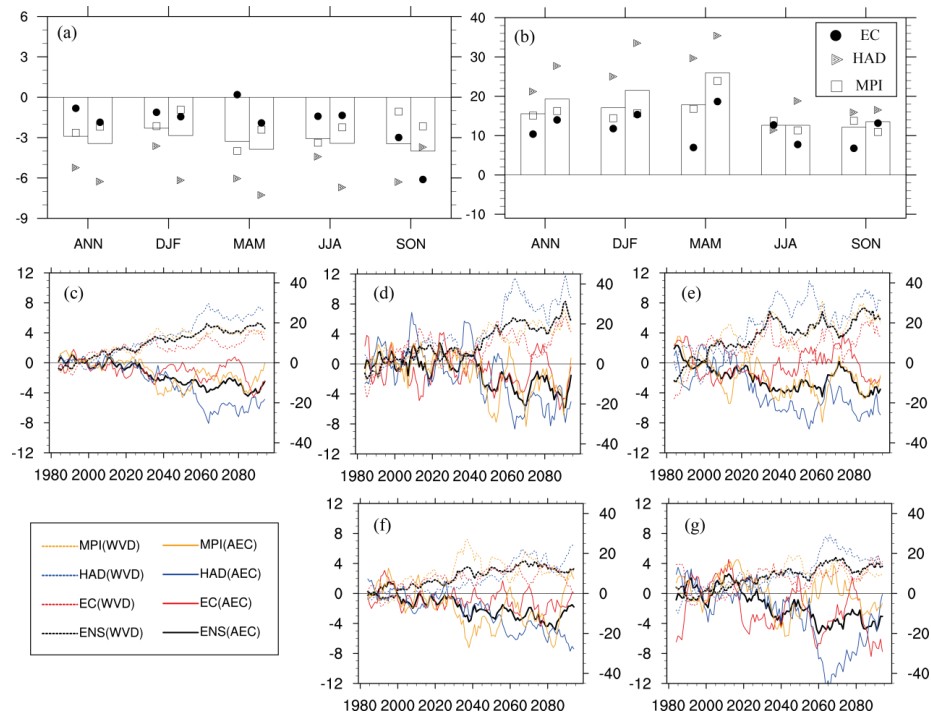


**Figure 6.** Same as Figure 5, but for Northeast China.



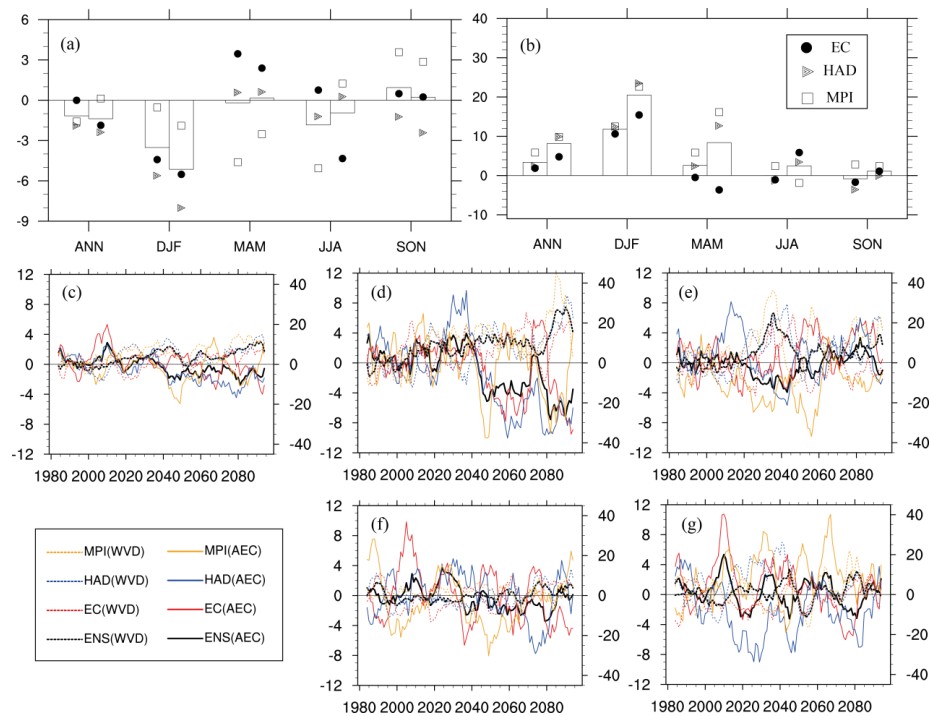


**Figure 7.** Same as Figure 5, but for Yangtze River Delta economic zone.





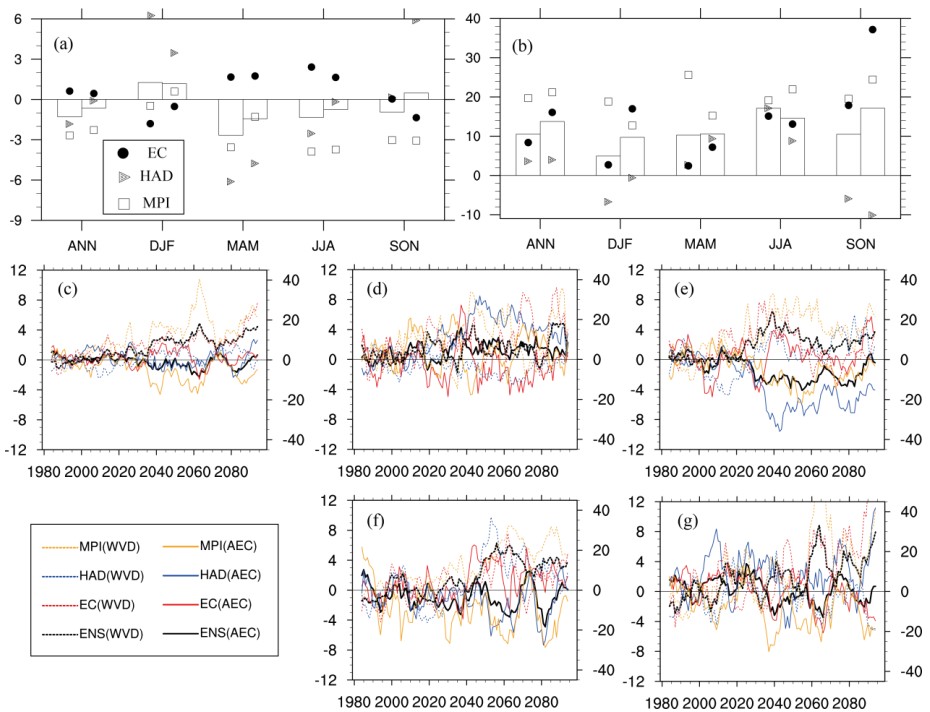


**Figure 8.** Same as Figure 5, but for Pearl River Delta economic zone.






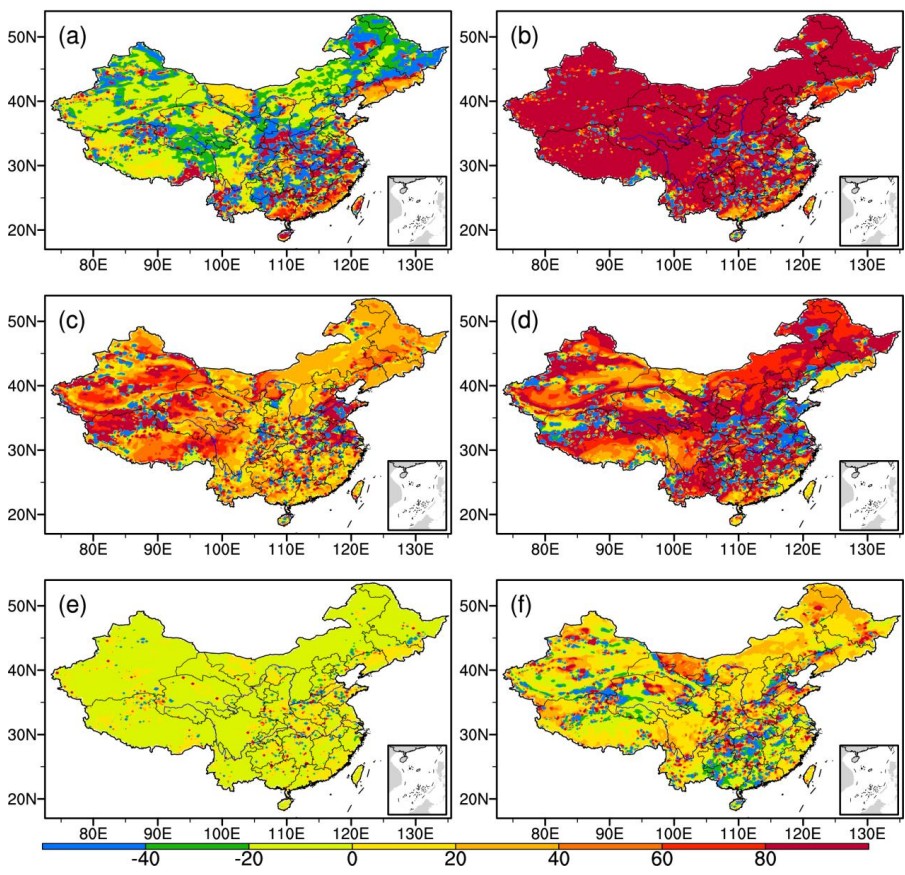

**Figure 9.** Relative contributions (unit: %) of individual components to annual AEC

change in the middle of the 21st century based on the ensemble results. (a)

precipitation, (b) ventilation, (c) wind speed averaged with the boundary layer, (d)

boundary layer depth, (e) nolinear term, (f) transient term.





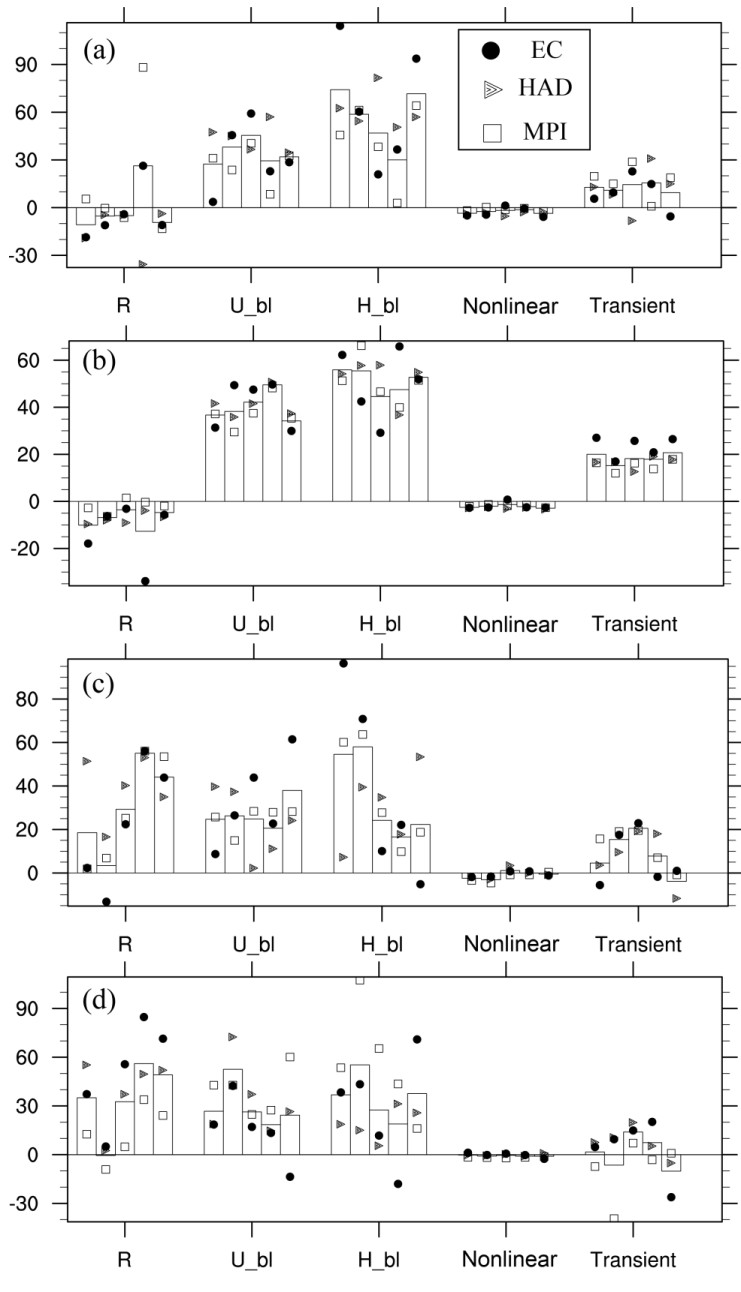


**Figure 10.** Relative contributions (unit: %) of individual components to annual AEC

change in the middle of the 21st century averaged over four main economic zones of

China: (a) BTH, (b) NEC, (c) YRD, (d) PRD. The bars represent the ensemble





projection and the marks represent the individual projection of the three members.
Bars from left to right in each group are in turn for annual, DJF, MAM, JJA, and
SON.