# Peer review of "Ensemble of Regional Climate Model Simulations"

_Atmospheric Chemistry and Physics, 2016_

## Referee Comment (RC1) · Anonymous Referee #2 · 27 Feb 2017

The authors examined the future haze pollution potential through downscaling three CMIP5 global models with the RCP4.5 scenario. The authors suggested that the air environment carrying capacity (AEC) tends to decrease and the annual weak ventilation days (WVD) tends to increase over large fraction of China in the future. Based on the AEC and WVD values, the probability of haze pollution risk over the Beijing-Tianjin-Hebei and Yangtze River Delta region increase in winter. Over Northeast China, the haze pollution risk will increase due to the decrease in AEC and increase in WVD.

1. My main criticism of this study is that the authors did not consider the impacts of reduced emissions in the future under RCP4.5 scenario on pollution risk. The haze pollution risk should relate to the aerosol concentrations. For example, higher background

concentration lead a higher pollution risk with the same AEC. Thus, a discussion about how the pollution risk changed due to changes in emission is at least needed.

2. Line 124- In the equation (1), how could authors distinguish the intensity of rainfall? The wet deposition with 10 mm/hour (and no precipitation in other 23 hours) should be different with 10 mm/24 hour.

3. Line 175-176 Why the AEC is underestimated over the southern Xinjiang and over-estimated over parts of North China? Which one is the major reason? Simulated precipitation, wind speed, or boundary layer depth?

4. Line 217- "Southwest China, northern North China, Northeast China…" A map is needed.

5. Line-304-306. Missing WVD bar in Fig.7b in JJA during the middle of the 21st century.

6. Line 364- Change "dominant" to "important". I don't think the annual 20%-30% could described as "dominate role".

7. Line 392-400. In addition to the wind speed and boundary layer depth, will the wind direction change in the future? Does it also have impact on the air environment carrying capacity and haze pollution potential?

---

## Author Comment (AC1) · 18 Apr 2017

**Reply to the Comments of the Referee #2**

**1. My main criticism of this study is that the authors did not consider the impacts of reduced emissions in the future under RCP4.5 scenario on pollution risk. The haze pollution risk should relate to the aerosol concentrations. For example, higher background concentration lead a higher pollution risk with the same AEC. Thus, a discussion about how the pollution risk changed due to changes in emission is at least needed.**

*Reply*: As pointed out by the Referee, the haze pollution risk does be related to the aerosol concentrations. However, what we discussed here focused on the atmospheric carrying capacity which is associated with wet deposition and ventilation and provides a condition to transport and dilute pollutants. It does not reflect real emission characteristics. Since there is no chemistry/aerosol module coupled in our experiments, the contribution of emissions to pollution change under RCP4.5 scenario cannot be calculated. In responding to the comment, we added a short discussion to address it in the manuscript.

**2. Line 124- In the equation (1), how could authors distinguish the intensity of rainfall? The wet deposition with 10 mm/hour (and no precipitation in other 23 hours) should be different with 10 mm/24 hour.**

*Reply*: We used 6-hourly data for the AEC calculation, so short-duration (no longer than 6 hours) and long-duration events can be roughly distinguished. Due to large volume of the outputs from ~120-yr simulations by regional climate model, the time resolution of the model output is limited especially for those 3D variables (e.g. geopotential height, wind speed).

**3. Line 175-176 Why the AEC is underestimated over the southern Xinjiang and overestimated over parts of North China? Which one is the major reason? Simulated precipitation, wind speed, or boundary layer depth?**

*Reply*: Similar to the contribution analysis in section 5, we applied the same method to investigate the contribution of different factors to the simulated AEC biases (Fig. S1). Overall, the simulation bias in boundary layer depth is the major factor for the simulated AEC bias over most parts of China (Fig. S1d).

[Figure]

Figure S1. Relative contributions (unit: %) of individual components to annual AEC biases based on the ensemble results. (a) precipitation, (b) ventilation, (c) wind speed averaged with the boundary layer, (d) boundary layer depth, (e) nonlinear term, (f) transient term.

**4. Line 217- "Southwest China, northern North China, Northeast China: : :" A map is needed.**

*Reply*: A map has been added in the revised manuscript (Fig. S2).

[Figure]

Figure S2. Four main economic zones of China, Beijing-Tianjin-Hebei region (BTH), Northeast China (NEC), Yangtze River Delta economic zone (YRD), and Pearl River Delta economic zone (PRD)

**5. Line-304-306. Missing WVD bar in Fig.7b in JJA during the middle of the 21$^{st}$ century.**

*Reply*: The percentage change of WVD in JJA during the middle of the 21$^{st}$ century is very small (0.008%), so the bar looks "missing".

**6. Line 364- Change "dominant" to "important". I don't think the annual 20%-30% could described as "dominate role".**

*Reply*: Changed.

**7. Line 392-400. In addition to the wind speed and boundary layer depth, will the wind direction change in the future? Does it also have impact on the air environment carrying capacity and haze pollution potential?**

*Reply*: As mentioned above, what we concerned in this study is the atmospheric carrying capacity that is only related to wet deposition and ventilation. The change of wind direction should be important. For example, the pollution from upwind emission sources could impact the air quality in some locations downwind. The wind direction may also change in the future. However, this topic is beyond the scope of this study. A short discussion has been added to clarify this issue in the manuscript.

---

## Referee Comment (RC2) · Anonymous Referee #1 · 25 May 2017

The authors examined the performance of the RegCM4 downscaling simulations on the air environment carrying capacity (AEC) and weak ventilation days (WVD) in China. Then, the AEC and WVD were projected for the period of 2046-2065 and 2080-2099 and some discussions were included.

General Comments: 1. The tile was "Projected Changes in Haze Pollution Potential in China", but what were analyzed were the AEC and WVD. Thus, the quantized relationships between haze pollution (days) and AEC, WVD should be proved and illustrated. That is, why the AEC and WVD could be used to represent the haze?

2. According to prior studies, the relative humidity was vital for the incident of haze. If you want to evaluate the haze pollution potential, the moisture conditions must be

considered.

3. "If each of the 6-hourly ventilation coefficients within one day is less than 6000 m2 s-1 , this day is counted as one weak ventilation day (WVD)". The threshold was cited from (Leung and Gustafson, 2005), a study of U.S. air quality, and was actually and firstly used by Pielke et al (1991). The question was that if the same threshold was reasonable for the recent haze pollution in China.

Pielke, R. A., R. A. Stocker, R. W. Arritt, and R. T. McNider (1991), A procedure to estimate worst-case air quality in complex terrain, Environ. Int., 17, 559– 574.

4. The recent winter haze pollution in North China or BTH area was severest, but the bias of historical estimations in winter and in North China was very significant. Thus, the error bars or confidence intervals must be discussed.

Specific Comments: 1. As well known, there were dozens of models in the CMIP5 project, so the reasons why only three models were selected should be supplemented. Furthermore, why did the authors only analyze two periods, i.e., 2046-2065 and 2080-2099?

2. The definition of Beijing-Tianjin-Hebei region (BTH), Northeast China (NEC), Yangtze River Delta economic zone (YRD), and Pearl River Delta economic zone (PRD) must be illustrated clearly.

3. In Figure 1–3, the resolutions of the observations was bad for evaluating the performance of Regcm4 downscaling. I noticed that the Era-interim used here was with the resolution 1.5*1.5o, and suggest that the data 0.5*0.5o should be better.

---

## Author Comment (AC2) · 21 Jun 2017

**Reply to the Comments of the Referee #1**

**General Comments:**

**1. The tile was "Projected Changes in Haze Pollution Potential in China", but what were analyzed were the AEC and WVD. Thus, the quantized relationships between haze pollution (days) and AEC, WVD should be proved and illustrated. That is, why the AEC and WVD could be used to represent the haze?**

*Reply*: As mentioned in the manuscript, the AEC, which is associated with the wet deposition and the ventilation, provides a direct way to investigate the change of the haze pollution potential, and has been applied in the operational work for the forecasting of pollution potential in China Meteorological Administration (CMA). According to previous studies, high (low) AEC is disadvantageous (advantageous) for the occurrence of haze pollution; longer (shorter) WVD corresponds to more (less) haze pollution incidents. This is the theory foundation for the relationships between haze days and AEC, WVD.

In respond to the comment, we carried out further analysis to verify the relationships of the haze days with the AEC and the WVD. The observed data of haze days, which are based on daily visibility and relative humidity records from ~2400 observation stations in China, are provided by the CMA. The occurrence of a haze day is defined with the criteria: 1) daily mean visibility below 10 km; 2) daily mean relative humidity less than 90%. Because the visibility data were collected in different forms before and after 1980 caused by different observational rules, the period 1980-2016 is used for analysis. As shown in Fig. S1, the haze mainly occurs in eastern China, particularly in the Beijing-Tianjin-Hebei region, the Yangtze River Delta, the Pearl River Delta, and Northeast China.

Correlations between annual haze days and AEC, WVD are calculated over each station. It shows that there are negative correlations between the haze days and the AEC, and positive correlations between the haze days and the WVD over most of stations, especially in eastern China where the haze mainly occurs (Figs. S2, S3).

Considering large uncertainties from emission sources and complex chemical process for the haze genesis, the relationships between haze days and AEC, WVD are quite robust and strong. The related analysis have been added in section 2.2.

[Figure]

Fig. S1.   Distribution of the averaged annual haze days over China during 1980-2016

[Figure]

Fig. S2.   Distribution of correlation coefficient between annual haze days and AEC

[Figure]

Fig. S3.    Distribution of correlation coefficient between annual haze days and WVD

**2. According to prior studies, the relative humidity was vital for the incident of haze. If you want to evaluate the haze pollution potential, the moisture conditions must be considered.**

*Reply*: What we focus on in this study is the atmospheric carrying capacity, which is only related to the wet deposition and the ventilation. The relative humidity does be an important factor affecting the incident of haze. However, it is beyond the scope of this study. In the manuscript, we added a short discussion to clarify this issue in the last paragraph.

**3. "If each of the 6-hourly ventilation coefficients within one day is less than 6000 m2 s-1 , this day is counted as one weak ventilation day (WVD)". The threshold was cited from (Leung and Gustafson, 2005), a study of U.S. air quality, and was actually and firstly used by Pielke et al (1991). The question was that if the same threshold was reasonable for the recent haze pollution in China.**

**Pielke, R. A., R. A. Stocker, R. W. Arritt, and R. T. McNider (1991), A procedure to estimate worst-case air quality in complex terrain, Environ. Int., 17, 559– 574.**

*Reply*: The threshold is just used to indicate the intensity of ventilation, similar to that

for precipitation or wind. The effect of ventilation on air pollutant may not change among different places. Thus, the value of less than 6000 m$^2$ s$^{-1}$ for ventilation coefficient was used not only in the U.S. (Leung and Gustafson, 2005; Trail et al., 2013), but also in other places such as India (Goyal and Rao, 2007; Manju et al., 2002), Athens (Kassomenos et al., 1995), and Thailand (Pimonsree, 2008).

Further, we conducted a sensitivity analysis to examine the relationships between WVD and haze days when different thresholds (3000, 5000, 6000, 7000, and 9000 m$^2$ s$^{-1}$) are used for the calculation of WVD. The result shows little change in their relationship under different thresholds (Fig. S4). Therefore, the threshold is reasonable for the analysis of this study.

Related clarification has been added in the second paragraph of section 2.2.

[Figure]

Fig. S4.   Probability density function on the distribution of correlation coefficient between annual haze days and WVD. Different thresholds are used for the WVD calculation. Two dash lines indicate the 95% confidence level

Goyal, S., and Rao, C. C.: Air assimilative capacity-based environment friendly siting of new industries—A case study of Kochi region, India, J. Environ. Manage., 84, 473-483, 2007.

Kassomenos, P., Kotroni, V., and Kallos, G.: Analysis of climatological and air quality observations from greater Athens area, Atmos. Environ., 29, 3671-3688, 1995.

Manju, N., Balakrishnan, R., and Mani, N.: Assimilative capacity and pollutant

dispersion studies for the industrial zone of Manali, Atmos. Environ., 36, 3461-3471, 2002.

Pimonsree, S.: PM10 dispersion during air pollution episode in Saraburi, Thailand, Asia-Pacific Journal of Science and Technology, 13, 1185-1190, 2008.

Trail, M., Tsimpidi, A., Liu, P., Tsigaridis, K., Hu, Y., Nenes, A., and Russell, A.: Downscaling a global climate model to simulate climate change over the US and the implication on regional and urban air quality, Geoscientific Model Development, 6, 1429, 2013.

**4. The recent winter haze pollution in North China or BTH area was severest, but the bias of historical estimations in winter and in North China was very significant. Thus, the error bars or confidence intervals must be discussed.**

*Reply*: Similar to the contribution analysis in section 5, we applied the same method to investigate the contribution of different factors to the simulated AEC biases (Fig. S6). Overall, the simulation bias in boundary layer depth is the major factor for the simulated AEC bias over most parts of China (Fig. S6d). The related discuss is added in the first paragraph of section 3.

[Figure]

Fig. S5. Relative contributions (unit: %) of individual components to annual AEC biases based on the ensemble results. (a) precipitation, (b) ventilation, (c) wind speed averaged with the boundary layer, (d) boundary layer depth, (e) nonlinear term, (f) transient term.

**Specific Comments:**

**1. As well known, there were dozens of models in the CMIP5 project, so the reasons why only three models were selected should be supplemented. Furthermore, why did the authors only analyze two periods, i.e., 2046-2065 and 2080-2099?**

*Reply*: In CMIP5, ~20 GCMs provide the six-hourly outputs of wind speed, temperature, and humidity for dynamical downscaling. However, to drive RCM

modeling, the ratio of the resolution between GCMs and RCMs should not exceed 6-8. So, only those GCMs with the resolution of 1~2 degree can be used to drive RegCM4 simulations. Due to the availability of CMIP5 GCMs and considering large volume of outputs for ~120-yr RegCM4 simulations, we just used these three GCMs for this study. This part has been added in section 2.1.

The periods 2046-2065 and 2080-2099 are commonly used to represent near-term and long-term in the CMIP5 projection, respectively (IPCC, 2013). This has been clarified in the first paragraph of section 4

**2. The definition of Beijing-Tianjin-Hebei region (BTH), Northeast China (NEC), Yangtze River Delta economic zone (YRD), and Pearl River Delta economic zone (PRD) must be illustrated clearly.**

*Reply*: A map has been added in the revised manuscript (Fig. S6, also see Fig. 1f in the manuscript).

[Figure]

Figure S6. Four main economic zones of China, Beijing-Tianjin-Hebei region (BTH), Northeast China (NEC), Yangtze River Delta economic zone (YRD), and Pearl River Delta economic zone (PRD)

**3. In Figure 1–3, the resolutions of the observations was bad for evaluating the performance of Regcm4 downscaling. I noticed that the Era-interim used here was with the resolution 1.5\*1.5$^{o}$, and suggest that the data 0.5\*0.5$^{o}$ should be**

**better.**

*Reply*: The native horizontal spatial resolution for the ERA-interim data is a T255 Gaussian grid, equivalent to a horizontal resolution of about 79 km or 0.75$^{\circ}$. The data with other resolutions are bilinear interpolated from the native Gaussian grid (https://software.ecmwf.int/wiki/display/CKB/). So in the revised manuscript, 0.75$^{\circ}$ *0.75$^{\circ}$ grid data are used. The conclusions for the evaluation are not changed (see Table 1, Table 2 and the figures for the observation in the manuscript).

---

## Author Response (AR2)

**Reply to the Comments of the Referee #2**

**1. Introduction: Haze in China is a very hot topic and raises a bunch of new studies recently. The authors may cite more recent papers to strengthen this part. Climate change (Cai et al., 2017, Nature Climate), Arctic sea ice loss (Zou et al., 2017, Science Advances) and decadal weakening of winds (Yang et al., 2016, JGR) suggested causes in climate view. Review article on light-absorbing aerosols (Qian 2015) and Asian monsoon-aerosol (Li 2016).**

**Cai W., et al. (2017), Weather conditions conducive to Beijing severe haze more frequent under climate change, Nat. Clim. Change, doi:10.1038/nclimate3249.**

**Y. Zou, et al. (2017), Arctic sea ice, Eurasia snow, and extreme winter haze in China, Sci. Adv., 3, p. e1602751**

**Yang, Y., et al. (2016), Increase in winter haze over eastern China in recent decades: Roles of variations in meteorological parameters and anthropogenic emissions, J. Geophys. Res. Atmos., 121, 13,050–13,065, doi:10.1002/2016JD025136.**

**Qian Y, TJ Yasunari, SJ Doherty, MG Flanner, WK Lau, J Ming, H Wang, M Wang, SG Warren, and R Zhang. 2015. "Light-absorbing Particles in Snow and Ice: Measurement and Modeling of Climatic and Hydrological Impact." Advances in Atmospheric Sciences 32(1):64-91. doi:10.1007/s00376-014-0010-0.**

**Li Z, W Lau, V Ramanathan, GX Wu, Y Ding, M Manoj, J Liu, Y Qian, J Li, T Zhou, J Fan, D Rosenfeld, Y Ming, Y Wang, J Huang, B Wang, X Xu, SS Lee, M Cribb, F Zhang, X Yang, C Zhao, T Takemura, K Wang, X Xia, Y Yin, H Zhang, J Guo, P Zhai, N Sugimoto, S Babu, and G Brasseur. 2016. "Aerosol and Monsoon Climate Interactions over Asia." Reviews of Geophysics 54(4):866-929. doi:10.1002/2015RG000500.**

*Reply*: The introduction part has been revised as suggestions.

[revised manuscript text omitted]